# Distinguishing ChatGPT(-3.5, -4)-generated and human-written papers through Japanese stylometric analysis

**Wataru Zaitsu**[1]*, **Mingzhe Jin**[2]

**1** Department of Psychological Counselling, Faculty of Psychology, Mejiro University, Tokyo, Japan,
**2** Institute of Interdisciplinary Research, Kyoto University of Advanced Science, Kyoto, Japan

* wataru0112csi@yahoo.co.jp

**Data Availability Statement:** All relevant data are within the Supporting Information files.

**Funding:** This work was partly supported by JSPS KAKENHI Grant Number JP22K12726. The funders did not participate in this study design, data

## Abstract

In the first half of 2023, text-generative artificial intelligence (AI), including ChatGPT from OpenAI, has attracted considerable attention worldwide. In this study, first, we compared Japanese stylometric features of texts generated by ChatGPT, equipped with GPT-3.5 and GPT-4, and those written by humans. In this work, we performed multi-dimensional scaling (MDS) to confirm the distributions of 216 texts of three classes (72 academic papers written by 36 single authors, 72 texts generated by GPT-3.5, and 72 texts generated by GPT-4 on the basis of the titles of the aforementioned papers) focusing on the following stylometric features: (1) bigrams of parts-of-speech, (2) bigram of postpositional particle words, (3) positioning of commas, and (4) rate of function words. MDS revealed distinct distributions at each stylometric feature of GPT (3.5 and 4) and human. Although GPT-4 is more powerful than GPT-3.5 because it has more parameters, both GPT (3.5 and 4) distributions are overlapping. These results indicate that although the number of parameters may increase in the future, GPT-generated texts may not be close to that written by humans in terms of stylometric features. Second, we verified the classification performance of random forest (RF) classifier for two classes (GPT and human) focusing on Japanese stylometric features. This study revealed the high performance of RF in each stylometric feature: The RF classifier focusing on the rate of function words achieved 98.1% accuracy. Furthermore the RF classifier focusing on all stylometric features reached 100% in terms of all performance indexes (accuracy, recall, precision, and F1 score). This study concluded that at this stage we human discriminate ChatGPT from human limited to Japanese language.

## Introduction

ChatGPT [1], released by OpenAI on November 30, 2022, can fluently generate human-like writings, such as essays, articles, academic papers, and news, and summarize sentences in English, Spanish, French, Japanese, among other languages. Based on generative pre-trained transformer (GPT) technology, this generative AI is a type of natural language processing (NLP) model known as large language model (LLM). GPT-3.5 is an early version of ChatGPT

collection, analysis, or decision to publish, except their role in paying the English proofreading and Publication Fee.

**Competing interests:** The authors have declared that no competing interests exist.

[1]. ChatGPT has attracted considerable attention worldwide and reached 100 million active users in two months since its release. Other chatbot systems, such as "Bard" by Google and "Bing AI" by Microsoft, have been released after ChatGPT. Although such chatbot systems can provide numerous benefits, they can also cause other problems. These chatbots can easily generate considerable fake news and exaggerate facts globally. In the educational sector, university students may directly use chatbots for writing reports, and the teachers may not detect this deception. Moreover, fake scientific papers, generated by chatbot without conducting research and experiments, could lead to various problems (i.e., plagiarism or academic fraud) in the scientific field. Most people cannot distinguish texts generated by AI and those written by humans [2, 3]. Therefore, OpenAI released an AI text classifier [4] on January 31, 2023, to detect AI-written text. However, according to OpenAI [4], the detection accuracy of this classifier is low for English texts. Though "specificity" can be evaluated (91% of human-written texts are correctly classified as "human-written"), "sensitivity" (true positive, the rate of identifying AI-generated text as "written by AI") reached only 26%. This result indicates that 74% of AI-generated texts are incorrectly classified as "human-written." Furthermore, the performance level of this classifier was lower for non-English languages. Therefore, verification for other language including Japanese is necessary.

Moreover, OpenAI released GPT-4 on March 14, 2023. Compared with GPT-3.5, this version achieved superior performance in terms of reasoning and conciseness. According to literature [5], the number of parameters of GPT-3.5 is 175 billion, whereas it is expected that GPT-4 has approximately 100 trillion parameters. Therefore, Study 1 investigated the following: (1) Differences in stylometric features between ChatGPT and human texts have been observed. If differences in stylometric features between ChatGPT and human texts exist, then we could classify both texts. (2) Whether stylometric features of GPT-4 generated texts are different from the earlier version (i.e., GPT-3.5) and more similar to human-written texts. Based on (1), a state-of-the-art study [6, 7] about GPT-3.5 reported the differences in parts-of-speech in English and Chinese texts produced by AI and humans. However, research involving the Japanese language is limited. Therefore, Study 1 compared three types of texts (GPT-3.5 and -4 generated, and human-written) by focusing on several Japanese stylometric features (parts-of-speech, etc.). Study 2 verified the classification performance between ChatGPT and humans considering several Japanese stylometric features using random forest (RF). Recent studies [6–10] have demonstrated that AI classifiers can detect AI-generated texts with high performance of approximately greater than 95%. Theocharopoulos et al. [9] compared classification performances among several classifiers, including classical machine learning (ML) such as logistic regression model, multinomial naive Bayes, and support vector machine. Furthermore, the best performances of deep learning methods (Bidirectional Encoder Representations from Transformers (BERT) and Long Short-Term Memory networks (LSTM)), attained an accuracy of 98.7%. On the other hand, Islam et al. [11] also reported that classical ML (Extremely Randomized Trees and RF) showed superior performances compared to neural classifier (LSTM). According to studies for authorship identification [12], Japanese stylometric analysis using RF achieved the best scores among several machine learning techniques (support vector machine, bagging, and boosting). Furthermore, the neural classifier results are incomprehensible (i.e., black box); however, RF classifier can calculate the understandable "importance" of variables (i.e., "whitebox" machine learning type) [13]. Therefore, we used RF as a classifier in this task.

Our study is novel because there is no research up to date for considering difference of stylometric features and verifying classification performance between AI and human for Japanese language.

## Method

### Sample

First, we gathered 72 Japanese academic papers for psychology (from "The Japanese Journal of Psychology" published by The Japanese Psychological Association, "The Japanese Journal of Criminal Psychology" published by The Japanese Association of Criminal Psychology, and "The Japanese Journal of Social Psychology" published by The Japanese Society of Social Psychology), written by 36 single authors. To control text lengths, texts containing approximately 1,000 characters were generated by randomly extracting the sentences from each paper, excluding citations. In case of authorship identification using stylometric analysis, texts containing more characters facilitate more accurate identification of authors; however, the minimum number of characters for valid level was determined as approximately 1,000 characters [14, 15].

Second, we generated 144 texts (72 texts of ChatGPT-3.5 and 72 texts of ChatGPT-4) of approximately 1,000 characters in Japanese. When generating texts, we instructed ChatGPT to generate Japanese papers with the same titles as each of the above academic paper: "Write 'Introduction (or Method, Results, Conclusion)' of a paper on 'the title of the paper'" to control these topics in each text.

### Stylometric features

We transformed the above Japanese text samples into four datasets corresponding to next four stylometric features, namely (1) bigrams of parts-of-speech, (2) bigrams of postpositional particle words, (3) positioning of commas, and (4) rate of function words. These stylometric features are efficient for classifying author and not dependent on topics. Bigram is an *N*-gram in case of *N* = 2, calculating the frequency of adjacent symbols (words, phrases, or characters) in each writing. Regarding the bigrams of parts-of-speech, we used Japanese POS tagger ChaSen [16] and counted the number of occurrences of "preposition + verb," "verb + adjective," and "noun + adjective" among others in each Japanese text. ChaSen could create more detailed Japanese parts-of-speech tags: for postpositional particle such as "case particle", "binding particle", and "ending particle", etc. The number of combinations for the bigrams of parts-of-speech was 955 (i.e., 955 variables in dataset). Jin [17] demonstrated the effectiveness of bigram of parts-of speech in identifying Japanese authors. In the case of bigram of postpositional particle words, we particularly counted the frequency of adjacent postpositional particle words such as "binding particle (は) + case particle (を)", "case particle (を) + case particle (が)", "binding particle (は) + case particle (の)" and the features reached to 533 variables. Third, positioning of commas [18] is used to identify authorship: "は +, (comma)", "が +, (comma)", and "を +, (comma)" (48 variables). Finally, the rate of function words denotes the percentage of no-meaning words excluding noun, verb, and adjective, such as auxiliary verb, conjunction, and postpositional particle in each text ("ある/ auxiliary verb", "さらに/ conjunction", and "に/ postpositional particle", the number of variables reached to 221). This study included adverbs as function words. Zaitsu & Jin [14] reported the validity of these four stylometric features in identifying Japanese authors by analyzing texts containing approximately 1,000 characters: the most effective features were the rate of function words; the next was bigrams of parts-of-speech. That study reported high-classification performance levels: 100% on sensitivity and 95.1% on specificity. Additionally, in this study, a single dataset was created to group four datasets, and the integrated dataset was analyzed to verify incremental validity using all stylometric features.

### Analysis procedure

In Study 1, we analyzed 216 texts of three classes (GPT-3.5, GPT-4, and Human) in five data-sets using classical MDS to examine (1) whether both AI-based distributions differ from those of human-written texts, and (2) whether distributions of GPT-4 generated text differ from those of GPT-3.5 and are closer to those by humans than of GPT-3.5. MDS can arrange objects into a dimension based on the similarity as distance between objects; therefore, the results of the configurations of objects in MDS are likely to depend on the distance. In this study, we used the symmetric Jensen-Shannon divergence distance ($d_{SJSD}$) as follows [19]:

$$d_{SJSD}(\boldsymbol{x}, \boldsymbol{y})^2 = \frac{1}{2} \sum_{i=1}^{n} \left( x_i \log \frac{2x_i}{x_i + y_i} + y_i \log \frac{2y_i}{x_i + y_i} \right)$$

In Study 2, we conducted leave-one-out cross validation (LOOCV) using RF to verify the accuracy of classification for 216 texts. In LOOCV, we constructed an RF classifier based on 215 texts (training set), except for a text (testing set) from all texts. Next, an RF classifier classified a text (testing set) into either of two classes (AI-generated or human-written). Same procedures were conducted against all texts one by one as "Testing set".

The R language was used for MDS (*cmdscale* function of the **stats** package) and RF (*randomForest* function of **random Forest** package). For RF, the number of decision trees was set to 1,000, and other hyperparameters were set to default.

## Results

### Study 1: Distributions of AI (GPT-3.5 and -4)-generated and human-written texts in each stylometric feature

Fig 1 displays the result of the distribution of texts concerning the bigrams of parts-of-speech. A part of distributions of three classes were mixed, but both AI-generated and human-written texts generally separated each other. Figs 2 (bigram of postpositional particle words) and 3 (positioning of commas) revealed similar results. These results indicate that a part of the human-written texts is undistinguishable from GPT-generated text. According to Fig 4, concerning the rate of function words, we can confirm few mixed texts. Fig 5, wherein all stylometric features are used, reveal complete discrimination of both AI and human. Figs 1–5 indicate that GPT-4 texts generally overlapped GPT-3.5 texts, and the distribution of GPT-4 were not close to human distributions compared with GPT-3.5.

### Study 2: Performance levels of classifying texts into two classes (AI-generated and human-written) by RF

We determined the confusion matrixes (Table 1) corresponding to four and all stylometric features and calculated four performance indexes: "Accuracy (the ratio of correct classifications among all texts)," "Recall (the ratio of correct outcomes out of all texts in each classified class)," "Precision (the ratio of correct results out of all texts in each true class)," and "F1 score (the value of harmonic mean based on combination of recall and precision,)." The details of confusion matrix and the equations for above performance metrics are described below.

$$Accuracy = \frac{a + d}{a + b + c + d}$$

$$Recall \; for \; GPT-generated = \frac{a}{a + b} \quad or \quad Recall \; for \; Human-written = \frac{d}{c + d}$$

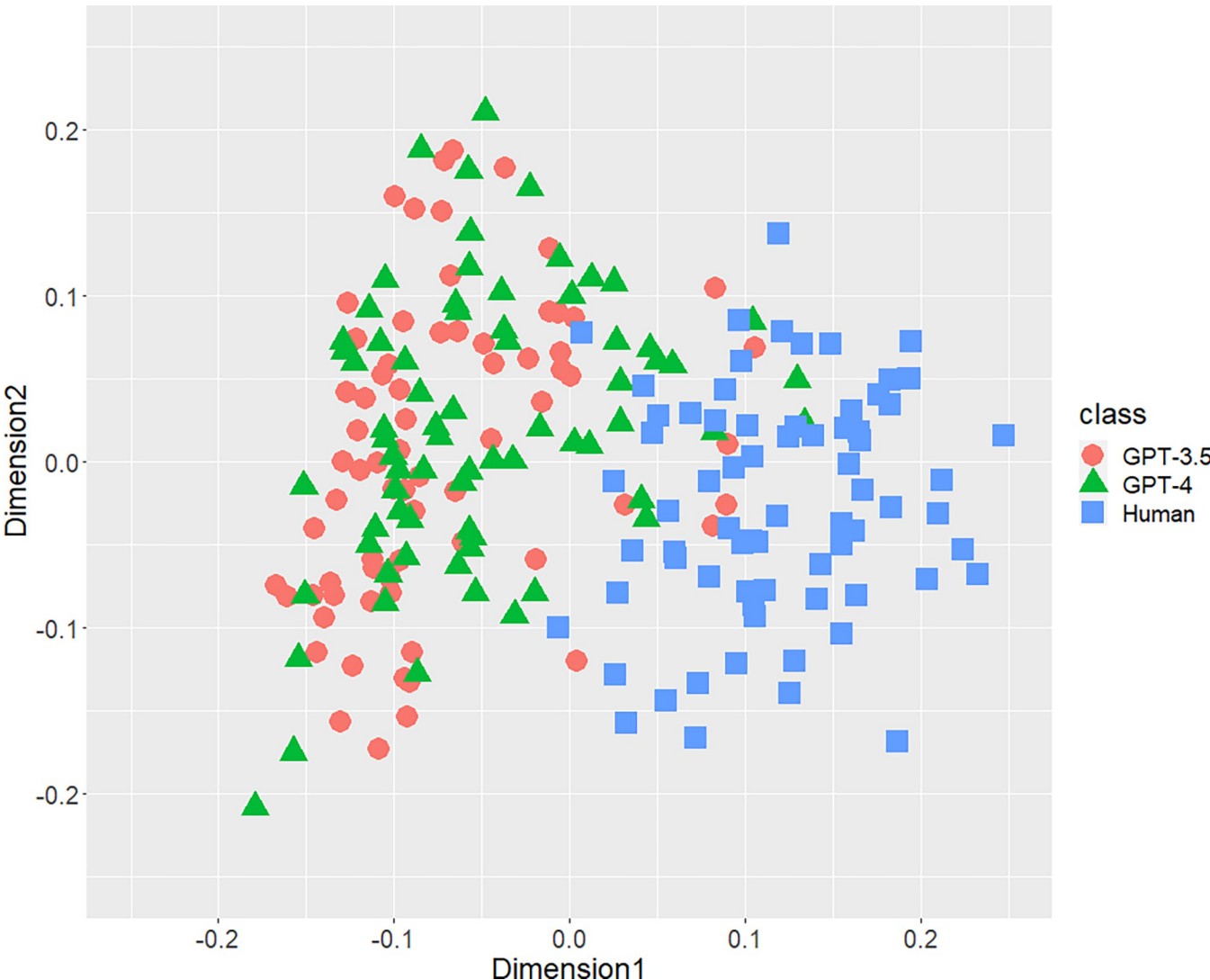

**Fig 1. Configuration of AI (GPT-3.5 and GPT-4)-generated and human-written texts, analyzed by MDS with $d_{SJSD}$, focusing on the bigrams of parts of speech.**

$$Precision\ for\ GPT{-}generated = \frac{a}{a+c} \quad or \quad Precision\ for\ Human{-}written = \frac{d}{b+d}$$

Tables 2–6 indicate that the rate of function words show the highest accuracy. By contrast, the bigram of postpositional particle words (Table 3) exhibited the lowest performance among four stylometric features. This feature also reached 100% in one side of recall or precision.

Additionally, we conducted 10-fold cross-validation for reference. Mean accuracies for each stylometric features are as follows: (1) the bigrams of parts-of-speech: 90.3% ($SD$ = 18.1%), (2) the bigram of postpositional particle words: 83.4% ($SD$ = 24.9%), (3) the positioning of commas: 91.2% ($SD$ = 13.2%), (4) the rate of function words: 94.9% ($SD$ = 11.1%), (5) all stylometric features: 96.3% ($SD$ = 7.4%).

The RF classifier can identify effective stylometric features for distinguishing GPT-generated from human-written texts as high "importance" of variables: "noun + postpositional particle" in bigrams of parts-of-speech, "case particle (が) + case particle (が)" in bigram of

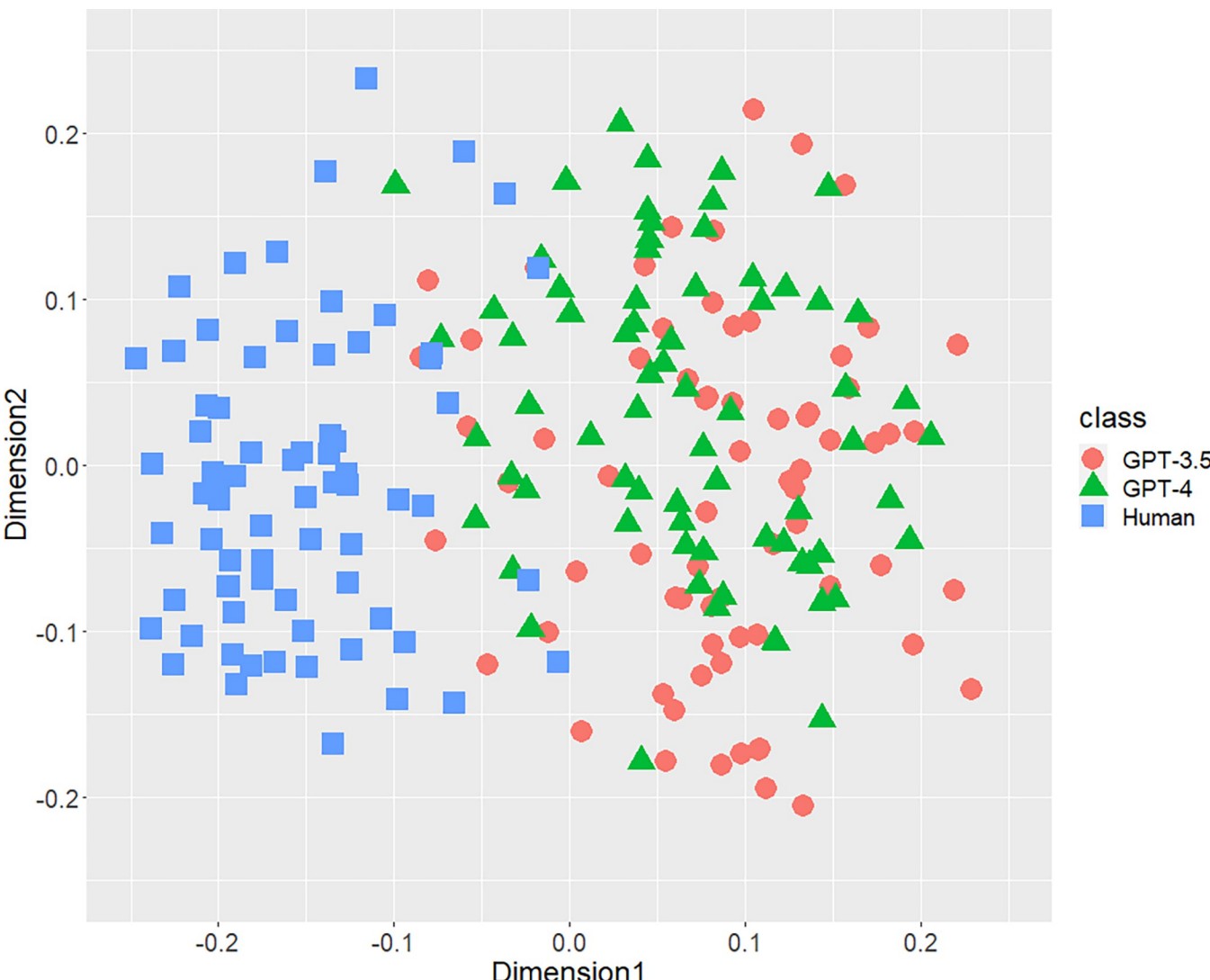

**Fig 2. Configuration of AI (GPT-3.5 and GPT-4)-generated and human-written texts, analyzed by MDS with $d_{SJSD}$, focusing on the bigram of postpositional particle words.**

postpositional particle words, "は +, (comma)" in the positioning of commas, and "本/ prefix" in the rate of function words.

## Discussion

According to MDS in Study 1, we first discovered the gaps in each stylometric feature between AI (GPT-3.5 and -4)-generated and human-written papers, particularly concerning the rate of function words among four features. These results indicated the obvious fact that AI and humans have a distinct writing process and mechanism. Furthermore, the distributions of both AI (GPT-3.5 and -4) showed considerable overlaps despite the number of parameters for both AI differing considerably. The distributions of GPT-4 were not closer to the distributions of humans compared with that of GPT-3.5. Thus, even if the number of parameters increases in the future, the distribution of ChatGPT-generated texts may not be close to that generated by humans in each stylometric feature. Furthermore, similar research approaches need to be

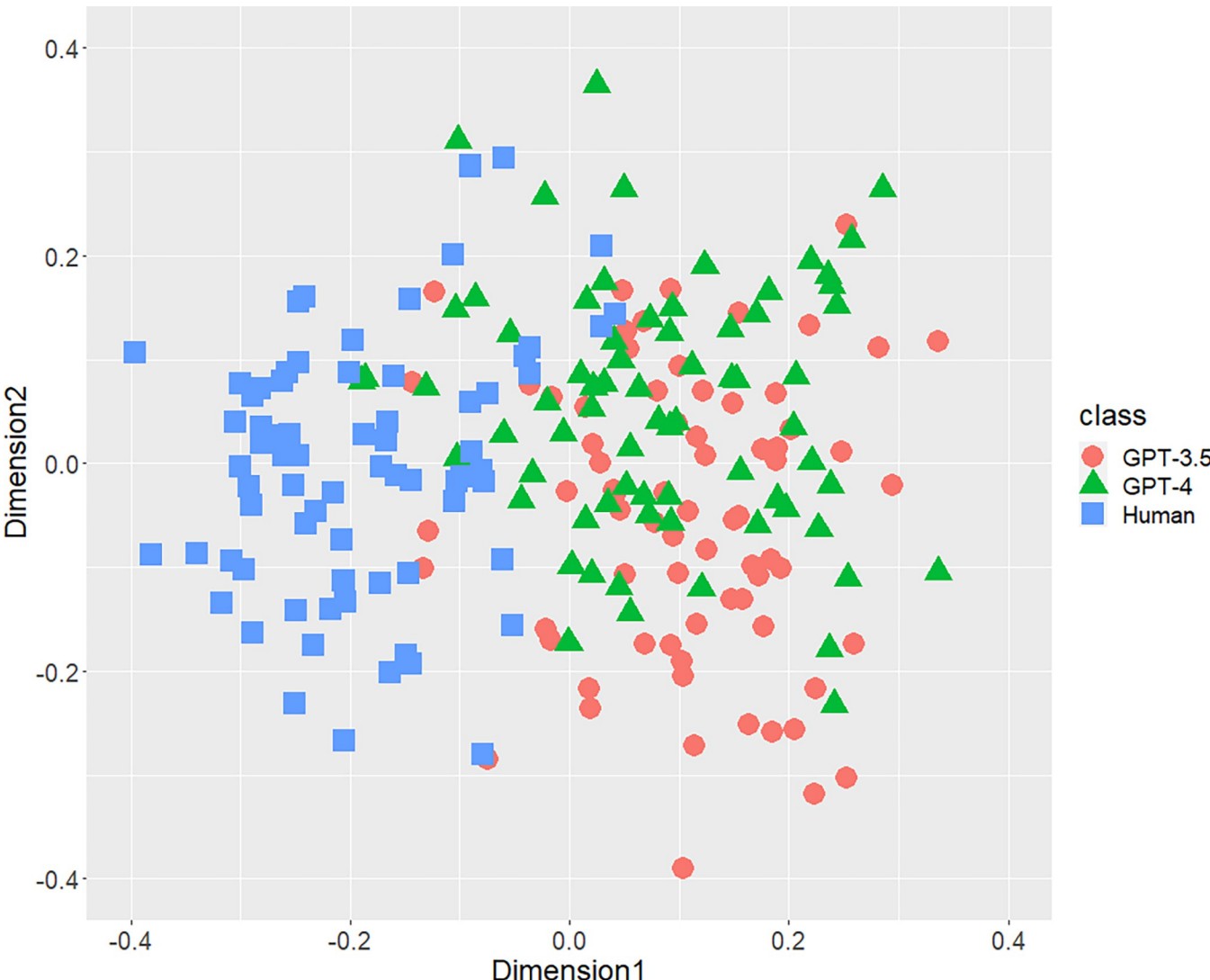

**Fig 3. Configuration of AI (GPT-3.5 and GPT-4)-generated and human-written texts, analyzed by MDS with $d_{SJSD}$, focusing on the positioning of commas.**

applied to the other chatbots. Second, in Study 2, we obtained high classification performance with the RF classifier focusing on several Japanese stylometric features. We could verify incremental validity because the RF for all stylometric features achieved higher performance (accuracy is 100%!) compared to that for each single stylometric feature. Summarizing the results, the further the distributions of texts of AI and human to each other in MDS, the better the classification performance will be by RF.

This current study has several limitations. First, the authors considered only Japanese language. According to the MMLU (massive multitask language understanding) benchmark [20], GPT-4 achieved the best performance in case of English, whereas its performance for Japanese language was ranked 16th. Japanese language is distinctive because of the co-existence of various forms (Kanji, Hiragana, and Katakana) in Japanese sentences and no space between words unlike English. Therefore, the analysis of English may yield distinct results from the current study. Second, academic papers exhibit lower flexibility because the writing has to be

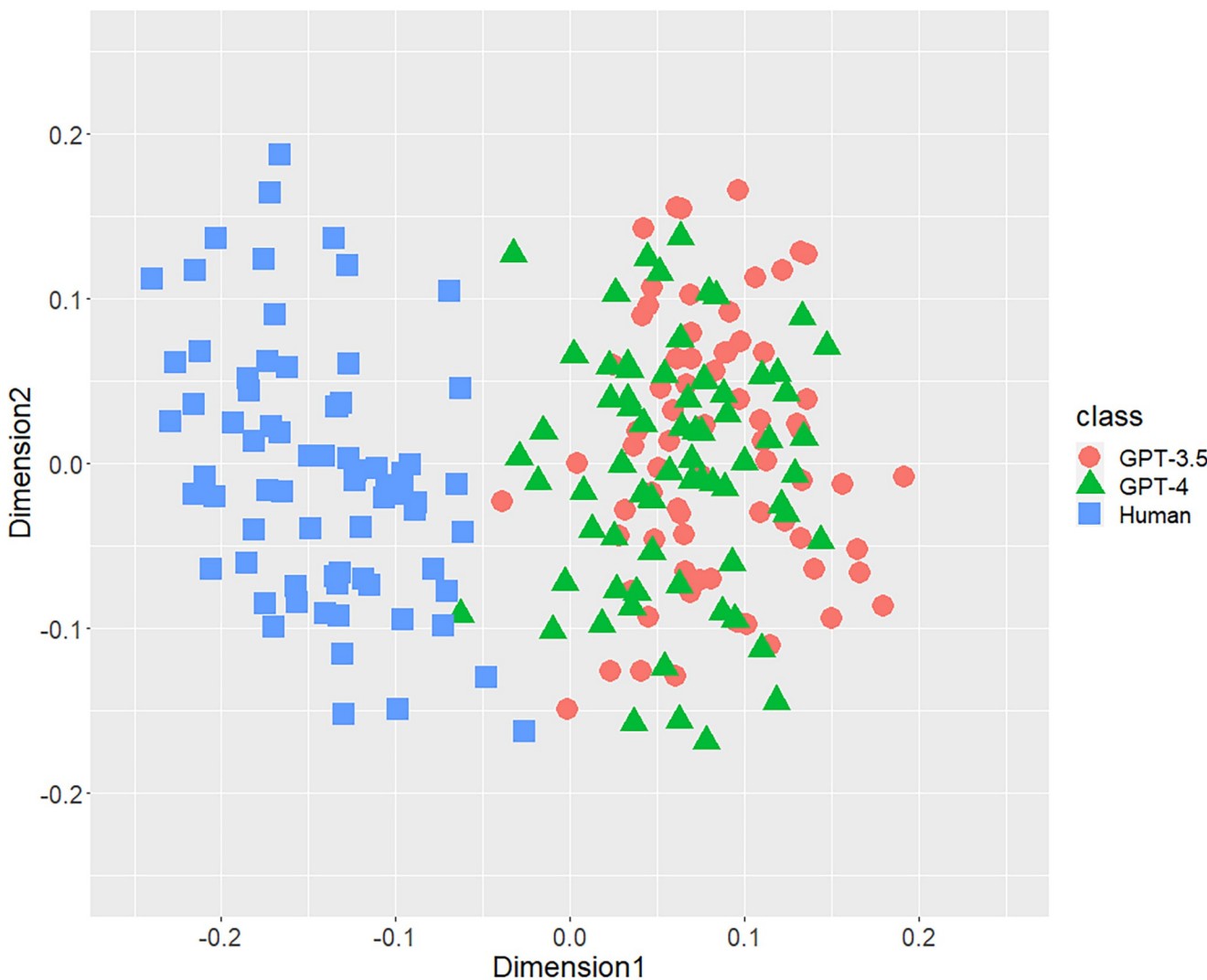

**Fig 4. Configuration of AI (GPT-3.5 and GPT-4)-generated and human-written texts, analyzed by MDS with $d_{SJSD}$, focusing on the rate of function words.**

according to the rules of each academic association. In future studies, materials with higher degrees of freedom, such as diary, blog, Twitter, should be used as samples. Third, other than ChatGPT, there are several chatbots ("Bard" by Google and "Bing AI" by Microsoft). Different AI systems may generate texts with different distributions of stylometric features. For such reason, it is necessary to study texts generated by other AI systems through the same methodology as this current study (i.e, stylometric analysis). Finally, texts containing approximately 1,000 characters were analyzed to control each text length. For texts containing more than 1,000 characters, higher performance levels compared to the current study may be obtained. However, for cases with fewer characters, the possibility of decreased performance levels exists.

Sophisticated text-generative AI will appear in the future. Predicting the future is difficult, and we may need to study AI to control generative AI.

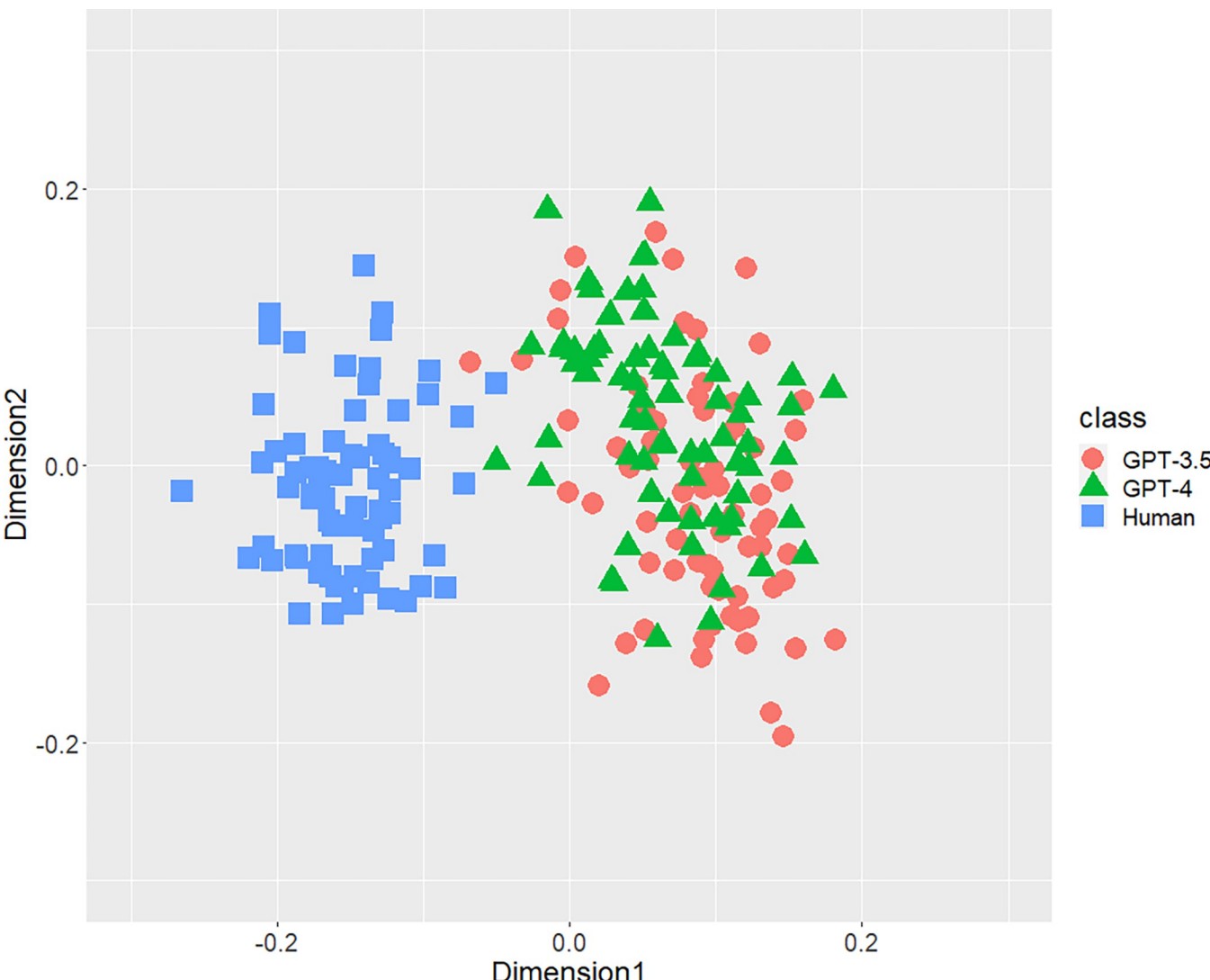

**Fig 5. Configuration of AI (GPT-3.5 and GPT-4)-generated and human-written texts, analyzed by MDS with $d_{SJSD}$, focusing on all stylometric features.**

## Conclusion

The conclusions of this study are as follows: (1) Stylometric features in Japanese were distinct between texts generated by ChatGPT (GPT-3.5 and -4) and texts written by humans. (2) Distribution of GPT-4 generated-texts overlapped the distribution of GPT-3.5 in stylometric features and were not close to those of humans. (3) Currently, we can classify AI-generated and human texts in the case of Japanese, with performance levels (accuracy, recall, precision, and F1 score) of 100%.

**Table 1. Confusion matrix.**

| True class | Classified class | |
|---|---|---|
| | **GTP-generated** | **Human-written** |
| GPT-generated | *a* | *b* |
| Human-written | *c* | *d* |

**Table 2. Confusion matrixes and performance indexes for the bigrams of parts-of-speech.**

| Accuracy | 95.4% | |
|---|---|---|
| Recall | 100.0% | 86.1% |
| Precision | 93.5% | 100.0% |
| F1 score | 96.6% | 92.5% |

**Table 3. Confusion matrixes and performance indexes for the bigram of postpositional particle words.**

| Accuracy | 92.6% | |
|---|---|---|
| Recall | 100.0% | 77.8% |
| Precision | 90.0% | 100.0% |
| F1 score | 94.7% | 87.5% |

**Table 4. Confusion matrixes and performance indexes for the positioning of commas.**

| Accuracy | 93.5% | |
|---|---|---|
| Recall | 98.6% | 83.3% |
| Precision | 92.2% | 96.8% |
| F1 score | 95.3% | 89.6% |

**Table 5. Confusion matrixes and performance indexes for the rate of function words.**

| Accuracy | 98.1% | |
|---|---|---|
| Recall | 100.0% | 94.4% |
| Precision | 97.3% | 100.0% |
| F1 score | 98.6% | 97.1% |

**Table 6. Confusion matrixes and performance indexes for all stylometric features.**

| Accuracy | 100.0% | |
|---|---|---|
| Recall | 100.0% | 100.0% |
| Precision | 100.0% | 100.0% |
| F1 score | 100.0% | 100.0% |

## Supporting information

**S1 Data.**
(CSV)

**S2 Data.**
(CSV)

**S3 Data.**
(CSV)

**S4 Data.**
(CSV)

**S5 Data.**
(CSV)

## Author Contributions

**Conceptualization:** Wataru Zaitsu.

**Data curation:** Wataru Zaitsu.

**Formal analysis:** Wataru Zaitsu, Mingzhe Jin.

**Funding acquisition:** Mingzhe Jin.

**Investigation:** Wataru Zaitsu.

**Methodology:** Wataru Zaitsu.

**Resources:** Mingzhe Jin.

**Software:** Mingzhe Jin.

**Supervision:** Mingzhe Jin.

**Visualization:** Wataru Zaitsu.

**Writing – original draft:** Wataru Zaitsu.

**Writing – review & editing:** Wataru Zaitsu.

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
