## [Decision Letter · Decision Letter 0]

23 May 2023

PONE-D-23-10817

Distinguishing ChatGPT(-3.5, -4)-generated and human-written papers through Japanese stylometric analysis

PLOS ONE

Dear Dr. Zaitsu,

Thank you for submitting your manuscript to PLOS ONE. After careful consideration, we feel that it has merit but does not fully meet PLOS ONE’s publication criteria as it currently stands. Therefore, we invite you to submit a revised version of the manuscript that addresses the points raised during the review process.

We look forward to receiving your revised manuscript.

Kind regards,

Michael Flor

Academic Editor

PLOS ONE

Journal Requirements:

   "This work was partly supported by JSPS KAKENHI Grant Number JP22K12726."

Additional Editor Comments:

In addition to comments provided by reviewer #1, please consider the following:

Line 78: “Bing” by Microsoft - maybe say “Bing AI” to differentiate it from the Bing search service.

Section “Stylometric features”, lines 157-162: what is the menaing of the term 'variables' in this section? What are 955 variables of part-of-speech bigrams? Are those 955 different types of ngrams? Please provide a clear explanation for all four features.

Also, please report which POS-tagger was used for this work.

Line 183: “This study reported...” It seems to refer to the Zaitsu&Jin 2018. In tha tcase you should write “That study”, because “This” might be interpreted as referring to the current manuscript.

Lines 309-312: “. Thus, even if the number of parameters increases in the future, the distribution of AI-generated texts may not be close to that generated by humans in each stylometric feature.”

This claim is unwarranted. It is based on comparing GPT 3.5 and 4, but the claims about the future are purely speculative. It is suggested to avoid such claim.

Line 333: what is LINE ? (explain in a footnote)

Please have a competent English speaker review the grammar and style of your manuscript. There are many grammatical and style errors. For example:

Line 31 “compared Japanese stylometric features generated by GPT”

Stylometric features were not generated by GPT. GPT generated just texts. Maybe you meant “stylometric features in texts generated by GPT”

Line 334-335: “there are several text-generated AI” - this is a very bad construction in English. Maybe you meant “chatbots”? “text-generated” literally means 'generated by text' and that does not make much sense.

Reviewers' comments:

Reviewer's Responses to Questions

**Comments to the Author**

1. Is the manuscript technically sound, and do the data support the conclusions?

Reviewer #1: Yes

Reviewer #2: Yes

2. Has the statistical analysis been performed appropriately and rigorously? 

Reviewer #1: Yes

Reviewer #2: Yes

3. Have the authors made all data underlying the findings in their manuscript fully available?

Reviewer #1: Yes

Reviewer #2: Yes

4. Is the manuscript presented in an intelligible fashion and written in standard English?

Reviewer #1: Yes

Reviewer #2: Yes

5. Review Comments to the Author

Reviewer #1: The authors investigate if and how well AI-generated Japanese texts can be distinguished from human-written Japanese texts. The sample of human-written texts was constructed by extracting segments of 1,000 characters from 72 scientific papers. The AI-generated sample consists of 2 x 72 texts generated by GPT-3 and GPT-4, respectively. The titles of the papers were used as prompts.

To classify the texts, the authors trained a random forrest classifier on 4 groups of stylometric features (bigrams of parts-of-speech, bigrams of postpositional particle words, positioning of commas and rate of function words). They show that based on each of the feature groups classification accuracies well over 90% can be achieved, with rate of function words being the most informative feature group. Combining all features, the proposed classifier labels all samples correctly.

The paper is well-written and easy follow, and the problem of recognizing AI-generated texts is timely and important. The presented study is clearly lacking in some aspects (see comments below), but the authors seem to be aware of the shortcomings, and the presented results constitute a valuable first step. Moreover, I think it is important to have studies on how well GPT can generate text in different languages and alphabets/writing systems since the majority of papers focuses on English and other languages using the Latin alphabet.

Some additional comments in no particular order:

- Overall the paper is well written, but there are occasional grammar mistakes ("an essays", "more close") or sentences that require some guess work from the reader ("we treated all texts as comprising approximately 1,000 characters"). Please have it proof-read again.

- The description of the technology underpinning GPT should be more precise - or removed if the authors are not familiar with it. For example, the authors write "This generative AI is based on generative pre-trained transformer (GPT) technology equipped with several natural language processing (NLP) models such as “transformer” architectures." GPT uses a transformer architecture for its underlying neural network. The network itself is the language model. It is not "equipped" with "several" models. A network architecture is not a model. Why put "transformer" in high commas? Also, the authors write that "GPT-4 has approximately 100 trillion parameters." To the best of my knowledge, OpenAI has not released any specifications for GPT-4, and estimates range from 400 billion to the 100 trillion the authors mention. However, these numbers have not been confirmed and cannot be verified.

- The authors should be careful comparing their detector to that of OpenAI. For example, the latter was evaluated on a "challenge set" that likely contains a large variety of samples, whereas the samples used in the paper are very homogeneous and narrow in scope.

- It would be interesting to see additional performance evaluations, for example, using 2/3/4-fold cross validation.

- How were the 1,000 character segments from the papers selected? Randomly? Why 1,000 characters? As the authors point out themselves, it would be interesting to see how the classification accuracy scales with the length of the segments.

- Please provide either code or all information necessary to reproduce the results. For example, which parameters were used for the random forest classifier? Which algorithm was used to generate the MDS plots?

- It would be interesting to see how well humans can tell the AI-generated samples from the human-written ones. Since the corpus is small, and the texts are short, such a study should be doable with a reasonable amount of time and effort. (This is probably too much to ask for in a revision.)

Reviewer #2: The stylometric features identified by the authors are quite commendable. However, I would like the authors to do a bit of literature survey, at least on the classifier part, and add more comparison of results with state-of-the-art models.

6. PLOS authors have the option to publish the peer review history of their article (what does this mean?). If published, this will include your full peer review and any attached files.

Reviewer #1: No

Reviewer #2: No

---

## [Author Response · Author response to Decision Letter 0]

2 Jun 2023

Answer and Response to Editor and Reviewer 1 comments:

Thank you for your constructive review.

I revised several points along with your comments.

Please confirm revised paper. 

Thank you.

1.”Bing by Microsoft”(p4.l78, p16.l371)

I modified “Bing” to “Bing AI”, along with reviewer’s comment.

2. Detailed explain: “Stylometric features” and “variables” (p8,l171)

 I more politely explained “Stylometric features”, “variables”, “POS tagger” in this paper.

3. revised representation (p9,l206)

 I replaced “this” to “that”, along with reviewer’s comment.

4. Increasing parameters and future (p15,l344)

 I revised some points, for example, “AI-generated” to “ChatGPT-generated” (Thus, even if the number of parameters increases in the future, the distribution of ChatGPT-generated texts may not be close to that generated by humans in each stylometric feature.). 

 But I think above claim is warranted, because this study showed evidence about relationship between difference of the number of parameters and no difference of stylometric features. So I used the expression 'may' because it is weak evidence.

5. Deleted “LINE”

 LINE is a kind of SNS mainly used in the Asian, but not around the world. So I deleted this word.

6. English Native check

 I requested an English proofreader.

7. revised representation

 I replaced “text-generated AI” to “chatbots”.

Answer and Response to Reviewer 1 comments:

1. Revised explanation of GPT technology in Introduction(p4,l71)

 I revised this paragraph in whole.

2. Additional analysis for cross validation(p12, l274)

 The number of K is too fine, it is not correct, because the sample size in this study is small. Therefore, I conducted “10-fold” cross validation.

3. Adding explain(p7, l155)

 I added explain of how to make texts (1,000 characters), along with reviewer’s comment.

4. Adding explain

 I explained hyperparameters for RF and used software and packages, along with reviewer’s comment.

Answer and Response to Reviewer 2 comments:

1. Adding reference about previous researches

 I added and wrote more references including state-of-the-art references.

---

## [Editor Report · Decision Letter 1]

22 Jun 2023

PONE-D-23-10817R1Distinguishing ChatGPT(-3.5, -4)-generated and human-written papers through Japanese stylometric analysisPLOS ONE

Dear Dr. Zaitsu,

Thank you for submitting your manuscript to PLOS ONE. After careful consideration, we feel that it has merit but does not fully meet PLOS ONE’s publication criteria as it currently stands. Therefore, we invite you to submit a revised version of the manuscript that addresses the points raised during the review process.

We look forward to receiving your revised manuscript.

Kind regards,

Michael Flor

Academic Editor

PLOS ONE

Journal Requirements:

Additional Editor Comments:

+++++++++++++++++++++++++++++++++++++++++

The revision R1 is adequate.

However, upon a close reading there are some additional aspects in the manuscript that need attention (mostly grammar, but not only).

Please attend to those, I hope this can be done quickly - this is good research and should be published quickly.

1.

Line 43-44: “both GPT ( -3.5 and -4 ) distributions are likely to overlap.” ==>

no need to put the hyphen before 3.5 and 4 in brackets.

Also they “are overlapping”, instead of “are likely to overlap”.

2.

Line 48 “random forest (RF)” ==> change to “random forest (RF) classifier”

3.

Line 83: “for writing reports without their inputs” ==> It is a bit unclear to whom 'their' refers. Also, there is always some input to a chatbot, so the sentence is confusing. Maybe just say “for writing reports”.

4.

Lines 103-105 – concerning the number of parameters in GPT 3.5 and 4, it is not enough to say “According to literature”, please provide a reference to the source.

5.

Line 106 and elsewhere in the manuscript: please convert expressions like “Research 1”, “Research 2” to “Study 1”, “Study 2”, etc., because this is the common terminology in academic English.

6.

Line 122 “high performances” ==> change to “high performance” (singular form)

7.

Line 136 “AI classifier results” ==> many things nowadays are called “AI classifiers”, so, to make you statement more clearer and accurate, may be say “neural classifier results” in this sentence.

8.

Line 269: “Table 1-4” ==> change to “Tables 1-4” (plural)

9.

Line 271: “exhibited the worst performance” ==> change to “exhibited the lowest performance” ('worst' invokes bad sentiment)

10.

Tables 1-4: the Precision, Recall and F1 scores are given per class. A common practice in reporting such statistics is to provide the overall Precision, Recall and F1 (per class is optional).

Please consider adding the overall values of Precision, Recall and F1 for each table. This is typically useful for authors who might want to cite your overall results in later studies.

11.

Line 341: “revealed overlapping” ==> change to 'show considerable overlaps'

12.

Line 371: “Different AI” ==> change to “Different AI systems”

13.

Lines 371-372: “may generate different distributions of stylometric texts” ==> change to “may generate texts with different distributions of stylometric features”

14.

Lines 372-373: “Generated texts should be verified using various generative AI” → this sentence is unclear and confusing. What did you intend to say? Did you mean that texts generated by other AI systems should be studied using stylometric features? Please clarify your message.

15.

Lines 386-387: “stylometric features in Japanese were distinct between AI (GPT-3.5 and 4 ) and human.” ==> change to “stylometric features in Japanese were distinct between texts generated by AI (GPT-3.5 and 4 ) and texts written by humans.”

16.

The manuscript uses the terms specificity, sensitivity, precision and recall; which may be confusing to many readers. Please consider including a short (!) footnote that explains those terms. You might find this page helpful: https://towardsdatascience.com/should-i-look-at-precision-recall-or-specificity-sensitivity-3946158aace1

17.

Finally, please address the following technical issue with your manuscript:

in your manuscript there are extra space characters within all words, so technically “word” is written as “w o r d”, etc. (see attached screenshot).

While this was not a problem for reading and reviewing, it should be fixed before the publication process.

+++++++++++++++++++++++++++++++++++++++++

---

## [Author Response · Author response to Decision Letter 1]

24 Jun 2023

Answer and Response to Editor:

Thank you for your constructive review.

We mostly revised grammar along with your whole comments.

In addition to grammar revision, we added explanation about confusion matrix and each performance metric (accuracy, recall, precision, and f1 score) along with your comments. 

Moreover, from the last time submission, we found new paper and added this paper.

Please confirm revised paper. 

Thank you.

Wataru Zaitsu

---

## [Editor Report · Decision Letter 2]

28 Jun 2023

Distinguishing ChatGPT(-3.5, -4)-generated and human-written papers through Japanese stylometric analysis

PONE-D-23-10817R2

Dear Dr. Zaitsu,

We’re pleased to inform you that your manuscript has been judged scientifically suitable for publication and will be formally accepted for publication once it meets all outstanding technical requirements.

Kind regards,

Michael Flor

Academic Editor

PLOS ONE

Additional Editor Comments:

Before publication, please revise the manuscript as follows:

In the first paragraph of the introduction section:

Line 57: "model known as large language model (LLP)" ==> change LLP to LLM

Line 58: "ChatGPT has attracted considerable attention worldwide and reached 10 billion active users in two months since its release".

This is rather embarrassing, since 10 billion is more that the current whole population on Earth.

I wonder how we missed this in the earlier rounds of review.

You probably meant 100 million. Please use this resource to cite this information:

https://www.reuters.com/technology/chatgpt-sets-record-fastest-growing-user-base-analyst-note-2023-02-01/

---

## [Editor Report · Acceptance letter]

3 Jul 2023

PONE-D-23-10817R2 

Distinguishing ChatGPT(-3.5, -4)-generated and human-written papers through Japanese stylometric analysis 

Dear Dr. Zaitsu:

I'm pleased to inform you that your manuscript has been deemed suitable for publication in PLOS ONE. Congratulations! Your manuscript is now with our production department. 

Kind regards, 

on behalf of

Dr. Michael Flor 

Academic Editor

PLOS ONE